# Socio-economic inequalities in burden of communicable and non-communicable diseases among older adults in India: Evidence from Longitudinal Ageing Study in India, 2017–18

Jhumki Kundu[☯], Ruchira Chakraborty[ID]*[☯]

International Institute for Population Sciences, Mumbai, India

☯ These authors contributed equally to this work.
* ruchirachakraborty71@gmail.com

## Abstract

Developing countries like India grapple with significant challenges due to the double burden of communicable and non-communicable disease in older adults. Examining the distribution of the burden of different communicable and non-communicable diseases among older adults can present proper evidence to policymakers to deal with health inequality. The present study aimed to determine socioeconomic inequality in the burden of communicable and noncommunicable diseases among older adults in India. This study used Longitudinal Ageing study in India (LASI), Wave 1, conducted during 2017–2018. Descriptive statistics along with bivariate analysis was used in the present study to reveal the initial results. Binary logistic regression analysis was used to estimate the association between the outcome variables (communicable and non-communicable disease) and the chosen set of separate explanatory variables. For measurement of socioeconomic inequality, concentration curve and concentration index along with state wise poor-rich ratio was calculated. Additionally, Wagstaff's decomposition of the concentration index approach was used to reveal the contribution of each explanatory variable to the measured health inequality (Communicable and non- communicable disease). The study finds the prevalence of communicable and non-communicable disease among older adults were 24.9% and 45.5% respectively. The prevalence of communicable disease was concentrated among the poor whereas the prevalence of NCDs was concentrated among the rich older adults, but the degree of inequality is greater in case of NCD. The CI for NCD is 0.094 whereas the CI for communicable disease is -0.043. Economic status, rural residence are common factors contributing inequality in both diseases; whereas BMI and living environment (house type, drinking water source and toilet facilities) have unique contribution in explaining inequality in NCD and communicable diseases respectively. This study significantly contributes in identifying the dichotomous concentration of disease prevalence and contributing socio- economic factors in the inequalities.

**Data Availability Statement:** The data is available on request from website of International Institute for Population Sciences, Mumbai. https://www.

iipsindia.ac.in/content/LASI-data. All the variable description is available in LASI India report available in the website: https://www.iipsindia.ac.in/content/lasi-publications.

**Funding:** The authors received no specific funding for this work.

**Competing interests:** The authors have declared that no competing interests exist.

## 1. Introduction

The goal of all health care systems is to improve, maintain, and restore community individuals' health [1]. The idea of health equality is rooted in principle of human rights [2], which states that there shouldn't be any systematic or possible disparities in one or more health-related areas among a population and socioeconomic subgroups. The concept encompasses equal access to services, financing, and health outcomes [3, 4]. A variety of demographic categories, including socioeconomic, gender, ethnic, geographic, and others, can be used to measure inequality. In this regard, Socio- economic inequality in health is one of the most common approaches.

The coexistence of communicable and non-communicable diseases (NCDs), also known as "the double burden of disease," has a significant impact on communities all over the world, but older people are particularly more affected [5]. Since the last decades of the 20th century, communicable disease related morbidity and mortality rates have decreased significantly on a global scale [6]. Due to the often-fragile healthcare systems, lack of funding, the majority of low-income and middle-income countries (LMICs) continue to suffer with a high communicable disease burden [7, 8]. Meanwhile, the burden of non-communicable diseases (NCD), such as diabetes, cancer, and cardiovascular illnesses, has increased enormously as a result of change in nutritional and lifestyle behavioural patterns (such as increasing fast food consumption, alcohol intake, and tobacco use) [9–11]. This double burden poses a serious threat to lower middle-income countries (LMIC), since the scarce financial resources are typically used to address the communicable disease problem while frequently ignoring the NCD problem [9, 12]. NCDs accounted for 61.8% of all deaths in 2016, according to the Lancet Global Burden of Disease Study [13], whereas communicable diseases were responsible for 27.5% of all fatalities. Many of the authors assume that health systems should give NCDs top priority when allocating resources. On the other hand, some other people contend that communicable diseases are quite common and worry that focus on NCDs will overshadow efforts to reduce disease burden and fatalities from infectious diseases like tuberculosis and diarrhoea [14]. India has reached the status of an ageing nation with more than 8% of its population being over 60 years of age [15, 16] with older population, growing faster relative to other parts of the world [17], challenges the healthcare system to some extent [18]. India is currently experiencing the double burden of communicable diseases and non-communicable diseases, with a minor drop in communicable diseases (CD) and a sharp increase in NCDs [19].

Examining the distribution of the burden of communicable and non-communicable diseases can present proper evidence to the policymakers to deal with health inequality. Worldwide numerous studies have focused on equity in health and how inequalities in terms of wealth or education or such socio-economic factors are co-related with health and health care access [20–24]. Previous researches focusing on socio-economic inequalities in relation to health in India are more concerned in maternal-child health or child nutrition [25, 26] Studies focusing older age groups, have explored inequalities in terms of self-rated health status [20, 27], multimorbidity, disability, or psychological distress [28–30] but no research from India had examined the extent of socioeconomic inequality in double burden of communicable and non-communicable disease separately. Therefore, as per the conceptual framework provided in Fig 1, the current study aims to explore the pattern of concentration of disease prevalence and contribution of different socio-economic and lifestyle factors in causing the inequality.

## 2. Materials and methods

### 2.1 Data source

The data for the study, has been taken from Longitudinal Ageing Study in India, (LASI wave I) as a cross-sectional analysis. It is a nationally representative data for adults aged 45 and above,

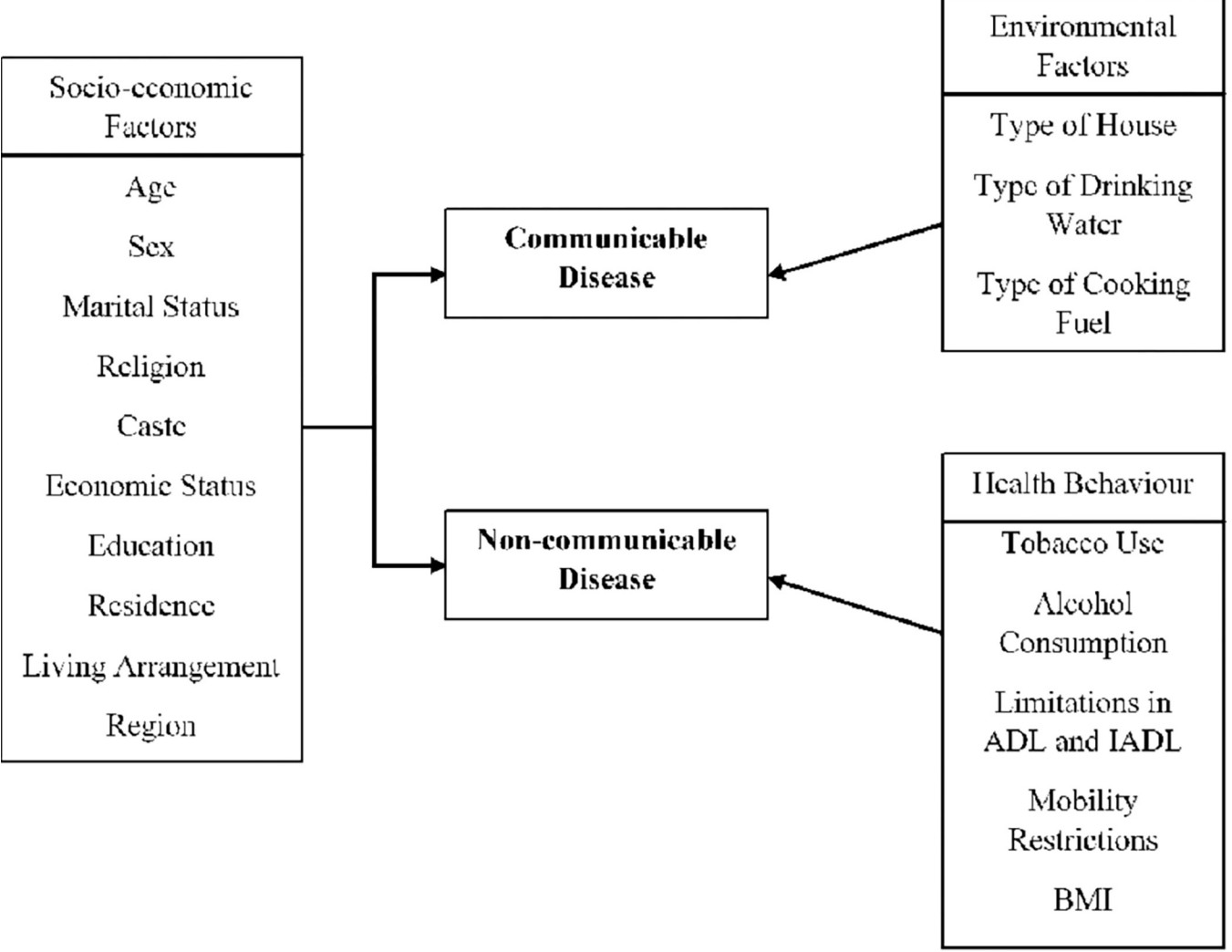

*Note: ADL- Activities in Daily Living; IADL- Instrumental Activities in Daily Living; BMI-Body Mass Index*

**Fig 1. Theoretical framework.**

conducted by International Institute for Population Sciences in collaboration with Harvard T. H Chan School for Public Health and University of Southern California in the 2017–2018. LASI is India's first comprehensive survey, which includes demographics, household economic status, chronic health conditions, functional health, mental health (cognition and depression), retirement for people aged 45 years and above. The survey covered 72,000 elderly age 45 and above and their spouses irrespective of age across all states and union territories of India [31]. In the study people aged 45 years are taken into consideration only to avoid younger spouse samples. Thus, the sample size included in the study is 65562, among them 30479 are males and 35083 are females. The survey adopted a multistage stratified area probability cluster sampling design to arrive at the eventual units of observation. Within each state, LASI

Wave 1 adopted three-stage sampling design in rural areas and four-stage sampling design in urban areas. In each state/UTs, the first stage involved selection of Primary Sampling Units (PSUs), that is, sub-districts (Tehsils/Talukas), and the second stage involved the selection of villages in rural areas and wards in urban areas in the selected PSUs. The present study is conducted on the eligible participant's age 45 years and above. The total sample size for the present study is 65562 (30479 males and 35083 females) older adults aged 45 years and above [31].

## 2.2 Variables

**2.2.1 Outcome variables.**   The outcome variables were binary in nature i.e. communicable diseases and non-communicable diseases were coded as "1" for having the disease and otherwise "0". The communicable disease included diarrhoea/gastroenteritis/ typhoid/ jaundice/ hepatitis, malaria/ chikungunya/ dengue and other infectious diseases like tuberculosis/ urinary tract infection and non- communicable disease includes Cardiovascular disease (CVD), Cancer, Chronic obstructive pulmonary diseases (COPD), Diabetes, and Bone disorder.

**2.2.2 Explanatory variables.**   Most of the explanatory and controlled variables are selected based on previous literatures [20, 32, 33].

To understand economic inequality, the monthly per capita expenditure (MPCE) was calculated utilizing information on household consumption. The sample households were surveyed using sets of 11 and 29 questions, on expenditure of food and non-food items respectively. The collection of non- food expenditure was based on reference periods of 30 days and 365 days, whereas the collection of food expenditure was based on a reference period of seven days. The 30-day reference period has been used to standardise both food and non-food expenditures. The summative indicator of consumption is the monthly per capita consumption expenditure (MPCE), which is calculated generating a weight by Principal component analysis. To understand prevalence of diseases according to socio-economic group the MPCE quintile is further binarized into poor and non-poor category. The set of explanatory variables only chosen for communicable diseases are:

- Type of house- recoded as pucca, semi pucca and kutcha [33].

- Type of toilet facility was recoded as unimproved and improved [34]. Pour-flush latrines, ventilated improved pit latrines, and pit latrines with a slab/covered pit are all examples of improved toilet facilities. Unimproved toilet facilities include pit latrines without slabs or open pits, bucket systems, shared facilities of any kind, no facilities (in a field or a bush), flush or pour-flush to elsewhere (i.e., not to a piped sewer system, septic tank, or pit latrine), and hanging toilets or hanging latrines.

- Source of drinking water was recoded as unimproved and improved [34]. Improved source of drinking water includes piped water, public tap/ standpipe, tube well or bore well, dug well, spring water and rain water. Unimproved water sources include tanker, cart with small tank, bottled water/pouch water, surface water and other sources of water.

- Cooking Fuel was recoded as clean fuel (LPG, Biogas and electricity) and unclean fuel (kerosene, charcoal, coal, crop residue, wood and cow dung etc.)

- On the other hand as non-communicable diseases are more related with sedentariness and health behaviour [20], the explanatory variables chosen only for NCD are:

- Body mass index was recoded as underweight, normal and overweight/obese. The participants having a body mass index (BMI) of 25 and above were categorized as obese/overweight whereas participant who had BMI as 18.4 and less were coded as underweight [35]. BMI is

calculated by dividing an individual's weight (in kilograms) by the square of their height (in metres).

- Physically Active was recoded as "Yes" and "No".

- Practicing yoga or pranayama was recoded as "Yes" and "No".

- Mobility restriction was recoded as "Yes" and "No".

Limitations in daily life activities or ADL- in bathing, eating, dressing or using toilet and limitation in instrumental activities of daily living or IADL- i.e., cooking, taking medicines or shopping groceries etc were classified into 2 separate sections; respondents do not have any issue and respondents having one and more limitation.

- Ever smoked or used smokeless tobacco was recoded as "Yes" and "No".

Ever consumed any alcohol beverages was recoded as "Yes" and "No" [36]

The control variables (Age, sex, place of residence, education, caste, religion, marital status and living arrangement and region) were selected to understand whether the prevalence and determinants of diseases vary in different socio-economic stratum.

## 2.3 Statistical analysis

Bivariate analysis was used in present study to identify the relationship between explanatory and outcome variables. Binary logistic regression analysis was used to estimate the association between the outcome variables (communicable and non-communicable) and other explanatory variables. The binary logistic regression model is usually put into a more compact form as follows:

$$\text{Logit } (Y) = \ln (p/1-p) = \alpha + b_1 X_1 + b_2 X_2 \ldots\ldots\ldots + b_k X_k$$

Where p is the probability of having the disease and α is the intercept, b is regression coefficients.

The socio-economic inequality in the burden of communicable and non-communicable disease among older adults in India have been examined using the poor-non-poor ratio and concentration Index. Ratio between the percentage of respondents with communicable and non-communicable disease in poorest MPCE quintile and the percentage of respondents with communicable and non- communicable disease in remaining quintiles are referred to as the "poornon-poor ratio." Higher value of the ratio indicates disease concentration is higher among poor and vice versa.

The concentration index is used to measure the overall inequalities in communicable and non- communicable diseases among the economic strata of older adults in India [37]. It is described as twice the area between the concentration curve, and the line of equality and the index is bounded between −1 and +1. There is no economic inequality in the distribution of diseases if the concentration index value is 0. Positive values of the concentration index show that diseases are disproportionately concentrated among rich people, while negative values indicate that diseases are concentrated among the poor.

Formally the concentration index is defined as,

$$C = -\frac{2}{\mu} \text{Cov}(h, r)$$

Here, concentration index is the covariance between health variable (h) and fractional rank in wealth status (r) rank distribution [38].

To understand the contribution of each of the socio-economic factor, decomposition analysis of the concentration index is done [39], where contribution of each individual social-economic,

demographic and health related factor to consumption-related inequality is decomposed. For any linear additive regression, keeping the health variable ($y_i$), i.e., NCD and communicable disease separately, as dependent variable, to a set of socio-economic independent variables ($x_{ki}$),

$$y_i = \alpha + \sum \beta_k x_{ki} + \varepsilon_i$$

Concentration index for $y_i$ can be calculated with following equation,

$$C = \sum \left(\frac{\beta_k \bar{x}_k}{\mu}\right) C_k + GC_\varepsilon / \mu$$

Here, $\mu$ is the mean of health variable ($y_i$); $GC_s$ is the generalised concentration index of the error term ($\varepsilon$); $\beta_k$ is the coefficient of the outcome variable.

The concentration index thus calculated has two components; the "explained" one which is the weighted sum (elasticity) of CI with the set of regressors; One unit change in explanatory variable is responsible for one unit change in dependent variable. And the other is the "residual" or say unexplained part ($GC_s/\mu$), indicates the prevailing socio-economic inequality in health outcome which cannot be explained through the chosen set of regressors ($x_{ki}$) [38]. Absolute contribution of each determinants is calculated by multiplying the elasticity and the CI of each control variable and the percentage contribution is calculated by dividing the absolute contribution with overall CI [20]. All the statistical analysis is done using Stata-16 software.

## 3. Results

### 3.1 Socio-demographic and economic profile of older adults in India

Table 1 presents the socio-demographic and economic profile of older adults in study sample. The percentage of poor and non-poor is almost similar for the age group of 60 to 69 (43.5%) and 70 and above (56.7%). 41.7% males belong to poor category whereas 42.5% female are poor. 49.2% of people who does not have any education are in poor category. 49.5% of the Scheduled caste are poor which increases to 58.5% for Scheduled Tribe. 46.3% of the widows and 47.5% of them who does not live with their spouse are poor. 57.5% of the poor have kutcha house type, 53.6% does not have improved toilet, 42.2% does not have improved source of drinking water and 50.8% does not use clean fuel. Among the poor, 53.3% are underweight. 50.1% people living in Central region are categorised as poor which is 31% in the north region.

### 3.2 Prevalence of communicable and non-communicable disease according to socio-economic strata

According to socio-economic strata, Fig 2 depicts the prevalence of communicable diseases among older adults in India. The graph demonstrates that among communicable diseases, diarrhoea/gastroenteritis affects older people more frequently, followed by malaria. The graph also demonstrates that, with the exception of typhoid, dengue fever, and tuberculosis, older people who were poor had higher percentages of all communicable diseases. Similarly, Fig 3 shows the percentage of non-communicable diseases among older persons. The graph shows that among the rich the percentage share of all non-communicable diseases is quite high.

### 3.3 State-wise inequalities in the prevalence of communicable and non-communicable disease

Fig 4 presents the poor-non-poor ratio for communicable and non-communicable diseases among older adults across the states in India. The findings significantly reveal that the

**Table 1. Socio demographic profile of the study population according to their economic category.**

| Background Characteristics | | Poor | | Non-poor | |
|---|---|---|---|---|---|
| | | Sample | Percentage | Sample | Percentage |
| **Age Group** | 45–59 | 13,170 | 40.5 | 20,928 | 59.5 |
| | 60–69 | 7,817 | 43.5 | 11,157 | 56.5 |
| | >70 | 5,144 | 43.3 | 7,346 | 56.7 |
| **Sex** | Male | 12,051 | 41.7 | 18,428 | 58.3 |
| | Female | 14,080 | 42.5 | 21,003 | 57.5 |
| **Residence** | Rural | 16,804 | 42.5 | 25,620 | 57.5 |
| | Urban | 9,327 | 41.2 | 13,811 | 58.8 |
| **Level of Education** | No Schooling | 14,712 | 49.2 | 16,106 | 50.8 |
| | less than 5years | 3,179 | 44.6 | 4,298 | 55.4 |
| | 5–10 years | 5,447 | 39.0 | 9,414 | 61.0 |
| | more than 9 years | 2,790 | 24.6 | 9,613 | 75.4 |
| **Religion** | Hindu | 19,486 | 42.2 | 28,613 | 57.8 |
| | Muslim | 3,447 | 43.8 | 4,356 | 56.2 |
| | Christian | 2, 417 | 43.3 | 4,119 | 56.7 |
| | Others | 779 | 31 | 2,341 | 69 |
| **Caste** | SC | 5,034 | 49.5 | 5,925 | 50.5 |
| | ST | 5,712 | 58.5 | 5,653 | 41.5 |
| | OBC | 10,049 | 40.9 | 14,580 | 59.1 |
| | Others | 5,298 | 32.6 | 13,209 | 67.4 |
| **Marital Status** | Currently Married | 18,926 | 40.8 | 29,843 | 59.2 |
| | Widowed | 6,335 | 46.3 | 8,257 | 53.7 |
| | Others | 868 | 40.8 | 1,330 | 59.2 |
| **Living Arrangement** | Living with spouse | 2,848 | 27.8 | 7,510 | 72.2 |
| | Living with spouse and children | 15,839 | 44.9 | 21,680 | 55.1 |
| | Living with Children | 5,570 | 47.5 | 6,871 | 52.5 |
| | Others | 1,874 | 40.6 | 3,370 | 59.4 |
| **Mobility Restriction** | No | 10,190 | 42.3 | 15,132 | 57.7 |
| | Yes | 15,832 | 42.1 | 24,192 | 57.9 |
| **ADL Restriction** | No | 22,221 | 42.1 | 33,519 | 57.9 |
| | Yes | 3,801 | 42.7 | 5,740 | 57.3 |
| **IADL Restriction** | No | 16,747 | 40.5 | 26,777 | 59.5 |
| | Yes | 9,231 | 45.1 | 12,467 | 54.9 |
| **Type of House** | Pucca | 11,022 | 34.3 | 23,770 | 65.7 |
| | Semi-pucca | 7,949 | 48.5 | 9,828 | 51.5 |
| | Kutcha | 6,419 | 57.5 | 5,181 | 42.5 |
| **Type of Toilet** | Not Improved | 6,881 | 53.5 | 5,630 | 46.5 |
| | Improved | 18,560 | 38.0 | 33,216 | 62.0 |
| **Source of Drinking Water** | Not Improved | 23,116 | 42.2 | 34,949 | 57.8 |
| | Improved | 2,793 | 42.4 | 4,150 | 57.6 |
| **Type of Fuel** | Not clean | 14,654 | 50.8 | 14,956 | 49.2 |
| | Clean | 10,786 | 34.3 | 23,889 | 65.7 |
| **BMI** | Underweight | 5,537 | 53.3 | 4,964 | 46.7 |
| | Normal | 12,865 | 43.3 | 18,469 | 56.7 |
| | Overweight | 3,994 | 32.5 | 8,858 | 67.5 |
| | Obese | 1,187 | 31.2 | 3,199 | 68.8 |
| **Region** | North | 3,437 | 31.9 | 8,529 | 68.1 |

*(Continued)*

**Table 1.** (Continued)

| Background Characteristics | | Poor | | Non-poor | |
|---|---|---|---|---|---|
| | | Sample | Percentage | Sample | Percentage |
| | Central | 4,733 | 50.1 | 4,174 | 49.9 |
| | East | 5,840 | 49.8 | 5,740 | 50.2 |
| | North-east | 3,181 | 42.0 | 5,332 | 58.0 |
| | West | 3,322 | 42.7 | 5,572 | 57.3 |
| | South | 5,618 | 32.6 | 10,084 | 67.4 |
| Total (N = 65,562) | | 26,131 | | 39,431 | |

SC: Scheduled Caste, ST: Scheduled Tribe, OBC: Other Backward Caste

prevalence of non-communicable disease was more concentrated in the non-poor strata whereas, communicable disease concentration is higher among the poor for older adults in India. The average poor non-poor ratio in India also advocates for the same. The pattern of concentration is similar throughout the states only the magnitude varies; e.g., Uttar Pradesh, Bihar, Jharkhand and Chhattisgarh have significantly higher concentration of communicable diseases among poor, higher than the national level.

### 3.4 Concentration curve

Fig 5 reveals that communicable disease was concentrated among older adults of poor socio-economic strata, as the curve is above the line of equality. The value of the concentration index

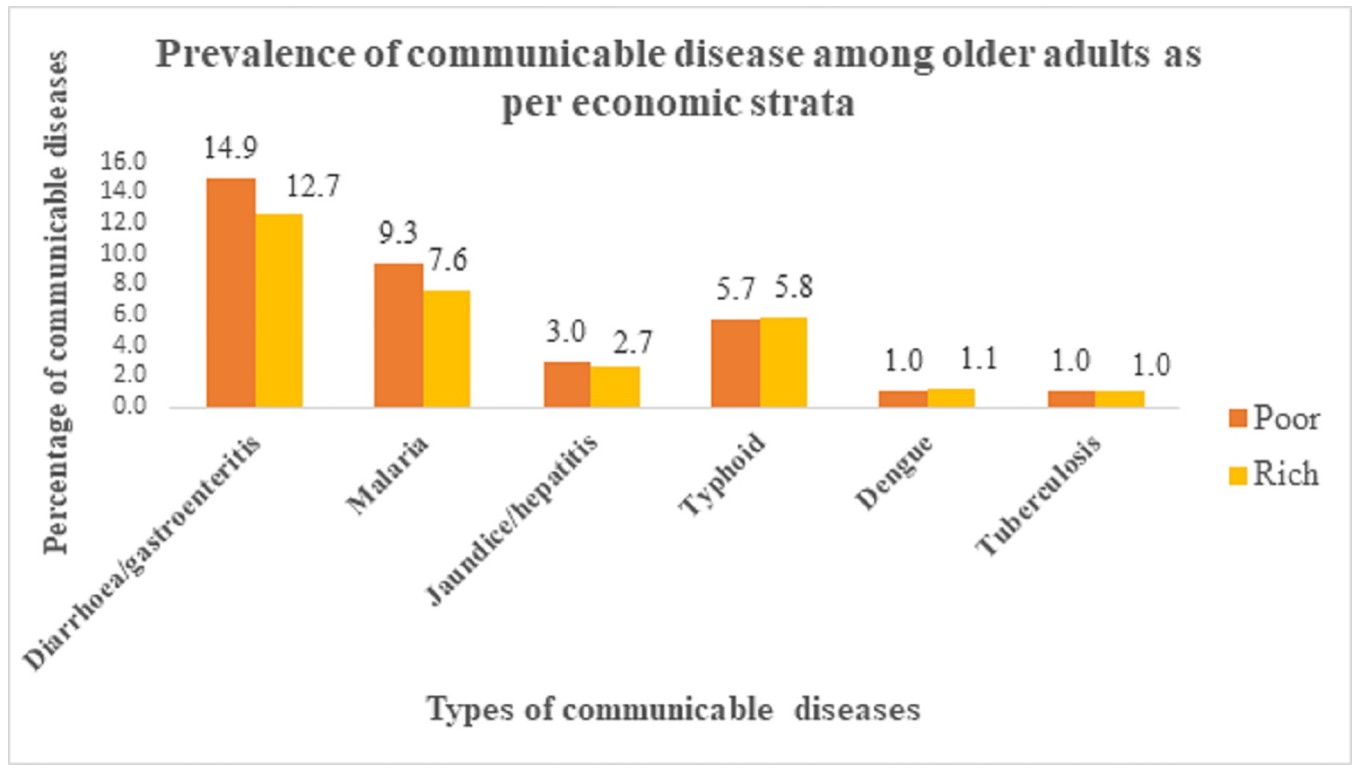

**Fig 2. Percentage of communicable diseases among older adults as per economic strata.**

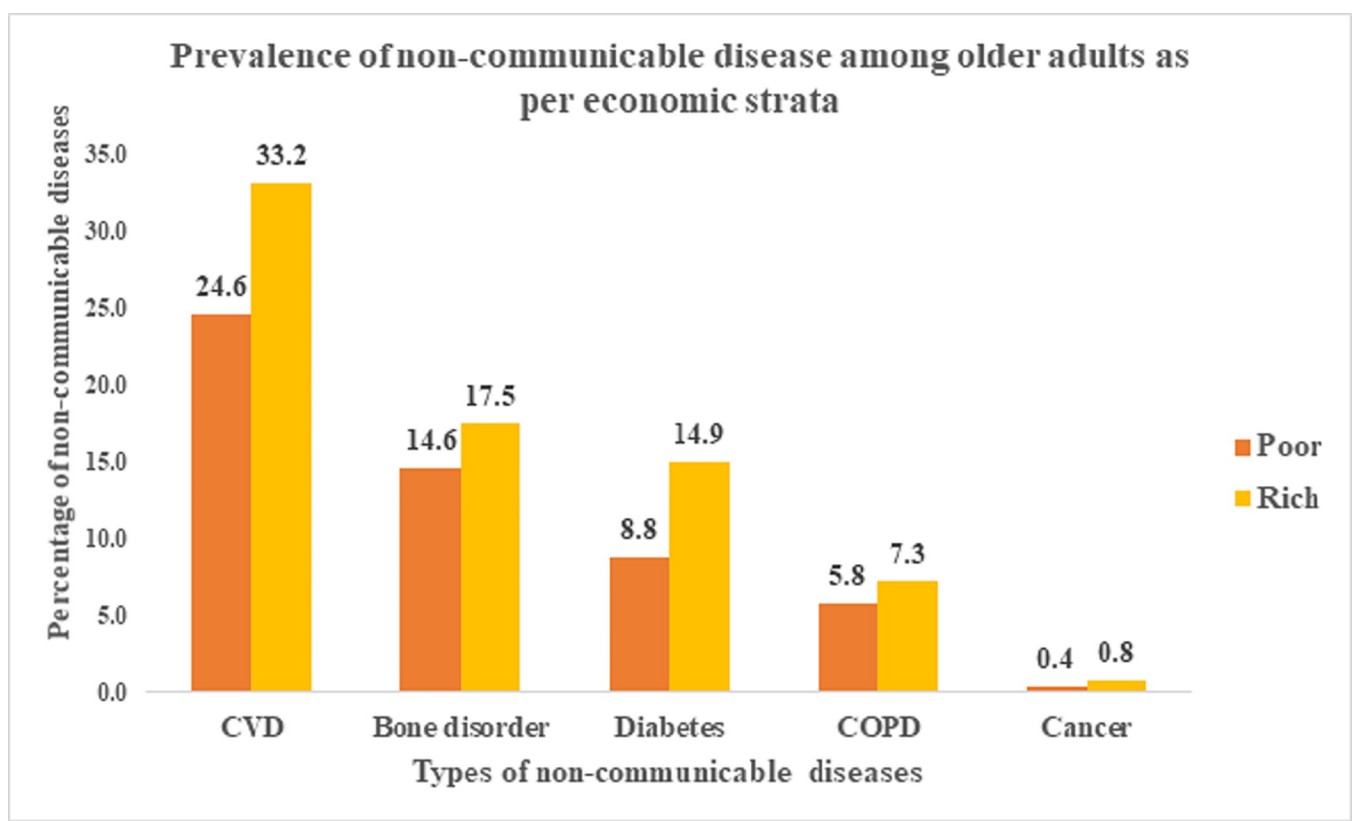

**Fig 3. Percentage of non-communicable diseases among older adults as per economic strata.**

is—0.043. But for non-communicable disease, the concentration is among socio-economically non-poor strata, where the curve situates under the line of equality. The value of the concentration index is 0.094. From the graph it also can be interpreted that the inequality is higher

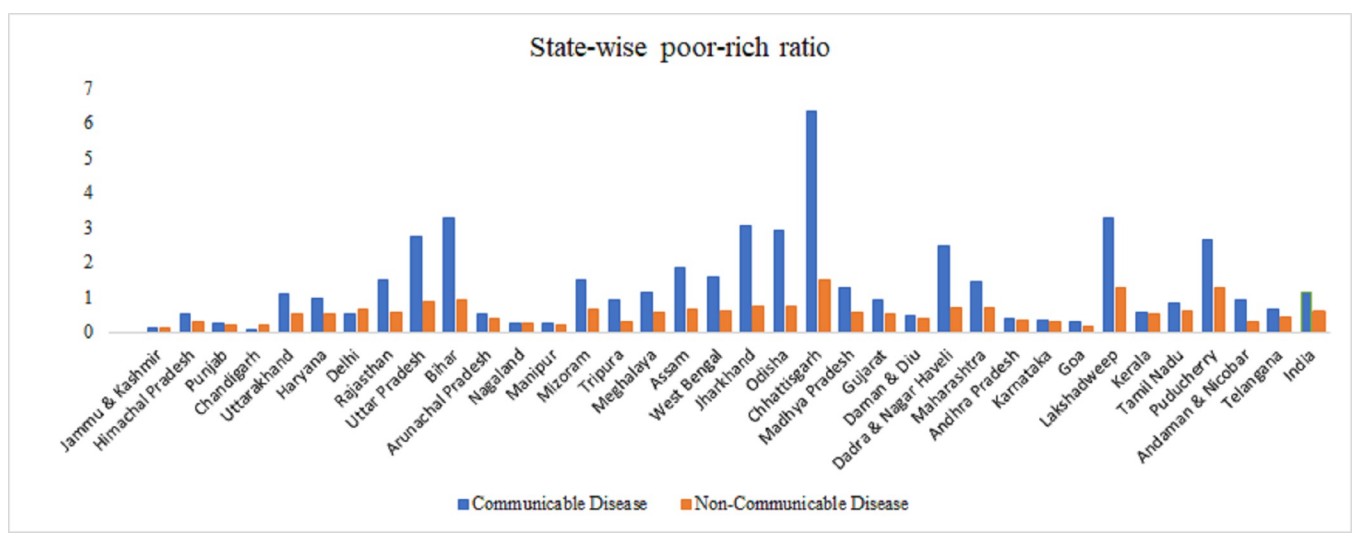

**Fig 4. State-wise poor-non-poor ratio in prevalence of communicable and non-communicable diseases.**

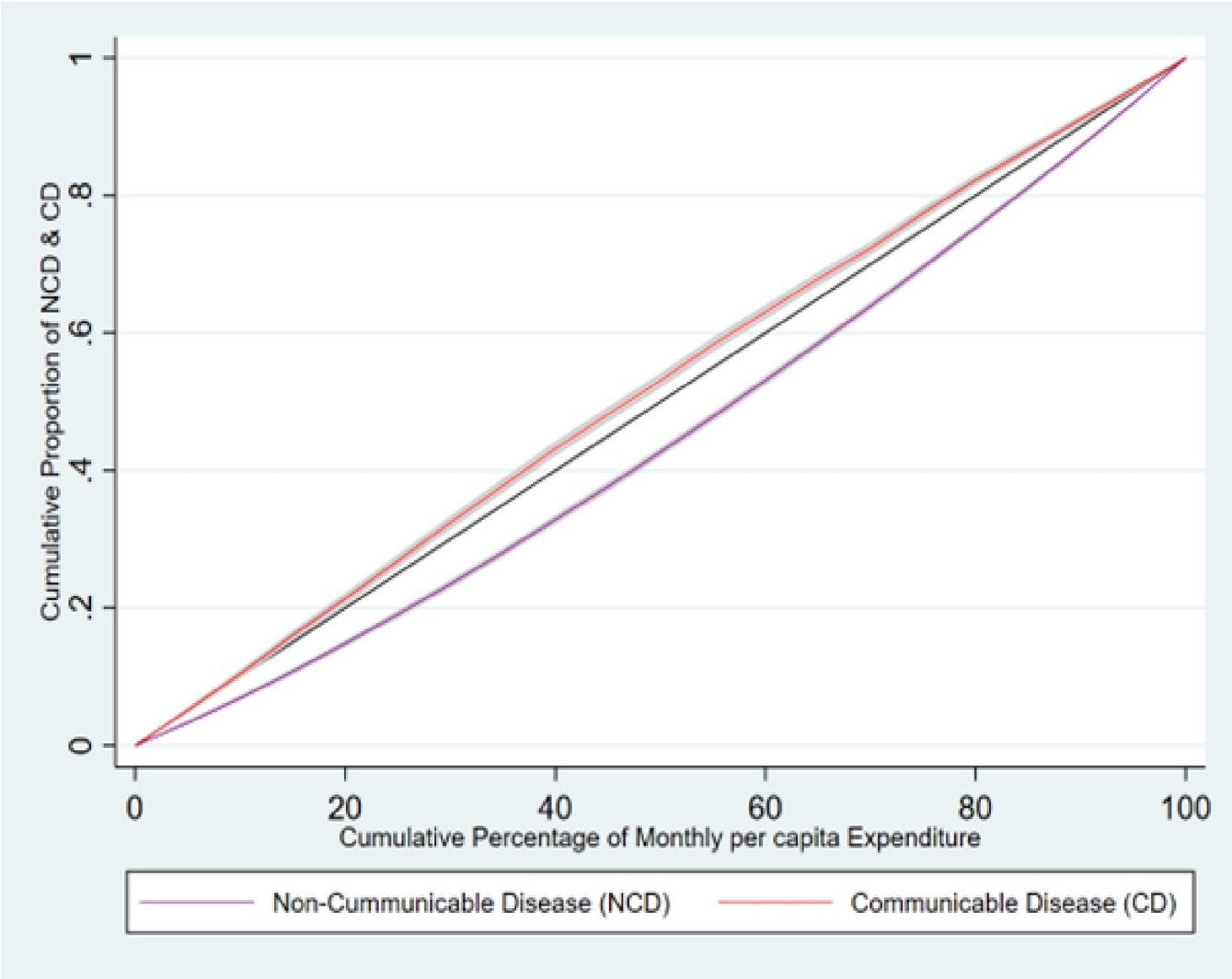

**Fig 5. Concentration curve for communicable and non-communicable disease among older adults in India.**

according to economic status in non-communicable disease as the area under the curve is more than the area under the curve of communicable disease.

### 3.5 Prevalence of communicable and non-communicable disease in different socio-economic background

Tables 2 and 3 shows how prevalence of communicable and non-communicable disease varies with different socio-demographic condition of older adults in India. Respondents were divided into two separate group of poor and non-poor to understand the variation. Weighted prevalence was calculated for separate sets of explanatory variables for both the group of disease. Poor people had a higher prevalence of communicable diseases across all age-groups than the non-poor people. The difference is also higher among male (2.2) than female (0.5) and in urban area (2.8) than rural (0.4). In case of communicable diseases, type of toilet facility and drinking water creates significant difference between poor and non-poor. In the north, central and north-east region the prevalence of communicable disease is higher among the poor

**Table 2. Prevalence of communicable disease according to socio-economic characteristics in India.**

| Background Characteristics | | Communicable Disease | | |
| --- | --- | --- | --- | --- |
| | | **Poor** | **Non-poor** | **Difference** |
| **Age Group** | 45–59 | 27.38 [26.18, 28.61] | 26.83[25.24, 28.48] | 0.6 |
| | 60–69 | 30.11 [28.50, 31.77] | 27.88 [26.40, 29.41] | 2.2 |
| | >70 | 29.41 [27.39, 31.51] | 27.90 [25.82, 30.08] | 1.5 |
| **Sex** | Male | 28.37 [27.09, 29.68] | 26.19 [24.61, 27.82] | 2.2 |
| | Female | 28.87 [27.66, 30.10] | 28.34 [27.02, 29.71] | 0.6 |
| **Residence** | Rural | 31.40 [30.36, 32.47] | 30.95 [30.09, 31.82] | 0.4 |
| | Urban | 22.42 [20.88, 24.03] | 19.61 [17.22, 22.24] | 2.8 |
| **Level of Education** | No Schooling | 31.18 [30.02, 32.36] | 31.01 [29.74, 32.31] | 0.2 |
| | less than 5years | 27.27 [24.80, 29.89] | 27.54 [25.15, 30.06] | -0.2 |
| | 5–9 years | 24.79 [22.89, 26.79] | 26.90 [25.16, 28.73] | -2.1 |
| | more than 10 years | 22.72 [20.21, 25.45] | 20.66 [17.81, 23.83] | 2.0 |
| **Religion** | Hindu | 28.92 [27.95, 29.91] | 28.41 [27.35, 29.49] | 0.5 |
| | Muslim | 29.15 [26.79, 31.63] | 23.25 [20.03, 26.81] | 5.9 |
| | Christian | 20.57 [17.31, 24.26] | 20.95 [17.99, 24.26] | -0.4 |
| | Others | 26.54 [21.96, 31.69] | 26.92 [24.31, 29.70] | -0.4 |
| **Caste** | SC | 30.41 [28.53, 32.35] | 29.17 [27.32, 31.08] | 1.2 |
| | ST | 32.55 [30.38, 34.79] | 31.76 [28.99, 34.66] | 0.8 |
| | OBC | 27.80 [26.45, 29.19] | 26.68 [24.81, 28.64] | 1.1 |
| | Others | 27.47 [25.43, 29.60] | 27.53 [26.28, 28.82] | 0 |
| **Marital Status** | Currently Married | 28.76 [27.75, 29.79] | 27.57 [26.41, 28.77] | 1.2 |
| | Widowed | 28.81 [26.96, 30.74] | 27.28 [25.21, 29.45] | 1.5 |
| | Others | 23.28 [18.90, 28.31] | 21.19 [15.13, 28.84] | 2.1 |
| **Living Arrangement** | Living with spouse | 26.16 [23.87, 28.57] | 27.58 [24.76, 30.59] | -1.4 |
| | Living with spouse and children | 29.31 [28.19, 30.47] | 27.65 [26.43, 28.91] | 1.6 |
| | Living with Children | 28.51 [26.57, 30.54] | 26.82 [24.48, 29.29] | 1.7 |
| | Others | 27.37 [23.99, 31.03] | 26.01 [22.80, 29.51] | 1.4 |
| **Type of House** | Pucca | 26.53 [25.26, 27.85] | 25.48 [24.10, 26.91] | 1.0 |
| | semi-pucca | 29.63 [28.10, 31.20] | 29.17 [27.80, 30.58] | 0.4 |
| | Kutcha | 31.62 [29.85, 33.45] | 34.71 [32.34, 37.16] | -3.1 |
| **Type of Toilet** | Improved | 26.02 [24.98, 27.09] | 24.94 [23.81, 26.11] | 1.1 |
| | Not improved | 33.91 [32.41, 35.45] | 36.94 [35.13, 38.78] | -3.0 |
| **Source of Drinking water** | Improved | 28.74 [27.86, 29.64] | 28.15 [27.10, 29.22] | 2.5 |
| | Not improved | 28.38 [24.15, 33.03] | 20.06 [17.88, 22.44] | 1.0 |
| **Type of Fuel** | Improved | 23.78 [22.49, 25.12] | 23.08 [21.69, 24.53] | 0.7 |
| | Not improved | 32.36 [31.23, 33.52] | 34.05 [32.84, 35.28] | -1.69 |
| **BMI** | Underweight | 34.20 [32.31, 36.14] | 34.89 [32.41, 37.45] | 2.5 |
| | Normal | 27.95 [26.78, 29.14] | 29.09 [27.62, 30.61] | -0.7 |
| | Overweight | 24.60 [22.46, 26.87] | 22.75 [20.77, 24.85] | -1.1 |
| | Obese | 26.95 [22.24, 32.25] | 22.81 [19.31, 26.74] | 4.2 |
| **Region** | North | 43.25 [41.10, 45.43] | 36.31 [34.89, 37.76] | 4.2 |
| | Central | 42.63 [40.66, 44.62] | 43.95 [41.84, 46.08] | 7.0 |
| | East | 28.34 [26.76, 29.98] | 27.09 [25.62, 28.61] | -1.4 |
| | North-east | 17.08 [15.14, 19.22] | 18.63 [17.08, 20.28] | 1.2 |
| | West | 21.69 [19.65, 23.89] | 23.20 [21.57, 24.93] | -1.5 |
| | South | 11.70 [10.01, 13.63] | 15.72 [13.14, 18.70] | -1.5 |

*(Continued)*

**Table 2.** (Continued)

| Background Characteristics | Communicable Disease | | |
|---|---|---|---|
| | Poor | Non-poor | Difference |
| Total (N = 65,562) | 26,131 | 39,431 | |

SC: Scheduled Caste, ST: Scheduled Tribe, OBC: Other Backward Caste; 95% confidence intervals are in brackets []

category; 27%, 43.3% and 28.3% respectively whereas in the east, west and south region the prevalence is higher in non-poor category; 44%, 18.6% and 23.2% respectively. But overall, in east region among the non-poor group, the prevalence of communicable disease is highest (44%). In case of non-communicable disease (Table 4) irrespective of other social and health factors the prevalence is always high among the non-poor. The poor non- poor difference is higher in the age group of 60–69 years (-11.8), among female (-9.5) and in rural areas (-9.3). Among all the categories of BMI, from being underweight to obese the prevalence is high among the non-poor category. Among the classifies regions, the prevalence of NCD is highest in the south, among non-poor people (55.6%).

### 3.6 Association of monthly per capita consumption expenditure with diseases

Table 4 shows the association of monthly per capita consumption expenditure with communicable and non-communicable diseases among older adults, which indicates, having higher wealth ensures lower odds of having communicable disease and vice versa. In the richest quintile the odds of having communicable disease is almost 70% less than the poorest. But in case of non-communicable disease the odds for the richest quintile increases up to 1.75 times than poorest.

### 3.7 Result of decomposition analysis

Tables 5 and 6 depicts the contribution of selected predictor variables to understand socio-economic inequality causing communicable and non-communicable diseases respectively. First model is capable of explaining 97.87% (CI = -0.047 out of CI = -0.046) of inequality in selected communicable disease whereas the other model explains 45.35% (CI = 0.078 out of CI = 0.0172) of inequality in NCD. For each factor a separate Concentration Index, sensitivity (elasticity), absolute and relative contribution to CI of Non-communicable and communicable disease is calculated. In case of NCDs, economic status (30.05%), rural residence (15.47%), being overweight (20.46%) and belonging to central region (10.23%) contributes the most in inequality. The CI value of East and North region also indicates very high inequality in NCD prevalence but with very low sensitivity they contribute little. On the other hand, age and mobility related restrictions has high sensitivity but does not point out much socio-economic inequality. Factors explaining inequalities regarding communicable disease are mostly region of residence; central region of India contributes the most (59.26%), and east and north region contributes around 30%, with very high sensitivity. Poor socio- economic status is the second most important factor contributing 25.71%, unimproved sanitary condition contributes 19.61% and rural residence contributes 15.69%. Exposure to unclean fuel, kutcha house type and unimproved source of drinking water has high sensitivity but contributes a little in causing inequality in communicable disease.

## 4. Discussion

The current study was an attempt to understand the socioeconomic inequalities in double burden of communicable and non-communicable diseases among the older population in India,

**Table 3. Prevalence of non-communicable disease according to socio-economic characteristics in India.**

| Background Characteristics | | Non-communicable Disease | | |
|---|---|---|---|---|
| | | Poor | Non-poor | Difference |
| **Age Group** | 45–59 | 33.45 [31.97, 34.96] | 41.93 [39.91, 43.98] | -8.4 |
| | 60–69 | 43.13 [41.36, 44.93] | 54.92 [53.02, 56.81] | -11.8 |
| | >70 | 51.47 [49.21, 53.71] | 59.24 [56.60, 61.83] | -7.7 |
| **Sex** | Male | 37.46 [35.95, 39.00] | 45.79 [43.99, 47.60] | -8.3 |
| | Female | 42.57 [41.18, 43.97] | 52.04 [50.16, 53.92] | -9.4 |
| **Residence** | Rural | 35.30 [34.18, 36.43] | 45.88 [44.31, 47.45] | -9.3 |
| | Urban | 51.39 [49.23, 53.54] | 50.13 [47.44, 52.82] | -7.5 |
| **Level of Education** | No Schooling | 37.89 [36.62, 39.18] | 51.18 [48.79, 53.56] | -8.0 |
| | less than 5years | 46.85 [43.92, 49.79] | 53.14 [49.12, 57.13] | -3.2 |
| | 5–9 years | 42.22 [39.81, 44.66] | 47.80 [46.64, 48.96] | -9.0 |
| | more than 10 years | 42.77 [39.09, 46.53] | 56.34 [50.10, 62.38] | -10.3 |
| **Religion** | Hindu | 38.64 [37.53, 39.77] | 50.52 [46.91, 54.11] | -9.2 |
| | Muslim | 48.04 [45.10, 50.99] | 53.65 [50.38, 56.90] | -8.3 |
| | Christian | 35.46 [31.13, 40.03] | 41.93 [39.91, 43.98] | -15.0 |
| | Others | 47.80 [42.19, 53.47] | 54.92 [53.02, 56.81] | -5.9 |
| **Caste** | SC | 39.31 [37.26, 41.40] | 46.04 [43.97, 48.12] | -6.7 |
| | ST | 23.31 [21.14, 25.64] | 34.29 [31.37, 37.35] | -11.0 |
| | OBC | 41.97 [40.28, 43.67] | 49.52 [47.00, 52.05] | -7.5 |
| | Others | 45.77 [43.57, 47.98] | 52.98 [51.53, 54.42] | -7.2 |
| **Marital Status** | Currently Married | 45.77 [43.57, 47.98] | 47.34 [45.91, 48.78] | -8.9 |
| | Widowed | 38.38 [37.20, 39.58] | 57.17 [54.47, 59.83] | -11.3 |
| | Others | 45.92 [43.81, 48.05] | 33.10 [23.85, 43.86] | 0.9 |
| **Living Arrangement** | Living with spouse | 33.98 [27.65, 40.94] | 50.53 [47.93, 53.12] | -8.1 |
| | Living with spouse and children | 42.41 [39.61, 45.25] | 46.19 [44.45, 47.94] | -8.6 |
| | Living with Children | 37.60 [36.29, 38.92] | 56.55 [53.29, 59.74] | -11.7 |
| | Others | 44.91 [42.63, 47.22] | 49.16 [44.37, 53.97] | -5.0 |
| **Mobility Restriction** | No | 44.21 [40.38, 48.12] | 33.76 [31.65, 35.94] | -8.5 |
| | Yes | 25.31 [23.69, 26.99] | 57.88 [56.43, 59.32] | -9.2 |
| **Restriction in ADL** | No | 48.72 [47.45, 49.99] | 46.10 [44.65, 47.55] | -9.5 |
| | Yes | 36.63 [35.54, 37.74] | 64.65 [61.88, 67.34] | -7.0 |
| **Restriction in IADL** | No | 57.74 [55.15, 60.29] | 43.97 [42.39, 45.55] | -9.1 |
| | Yes | 34.91 [33.65, 36.20] | 58.67 [56.61, 60.71] | -10.1 |
| **Tobacco Consumption** | No | 48.22 [46.53, 49.91] | 52.02 [50.15, 53.89] | -9.5 |
| | Yes | 42.49 [41.12, 43.87] | 43.90 [42.54, 45.27] | -7.2 |
| **Alcohol Consumption** | No | 36.66 [35.12, 38.23] | 50.22 [48.80, 51.63] | -8.7 |
| | Yes | 41.50 [40.41, 42.60] | 38.35 [35.76, 41.01] | -12.0 |
| **BMI** | Underweight | 26.35 [23.50, 29.41] | 35.34 [32.88, 37.87] | -3.7 |
| | Normal | 31.60 [29.68, 33.59] | 45.05 [43.60, 46.50] | -7.5 |
| | Overweight | 37.57 [36.21, 38.94] | 58.14 [54.43, 61.75] | -4.4 |
| | Obese | 53.65 [50.66, 56.61] | 70.51 [65.74, 74.86] | -5.3 |
| **Region** | North | 65.17 [59.11, 70.77] | 51.56 [50.07, 53.04] | -5.9 |
| | Central | 45.68 [43.53, 47.85] | 36.62 [34.66, 38.62] | -7.3 |
| | East | 29.31 [27.43, 31.26] | 49.15 [47.31, 50.98] | -10.3 |
| | North-east | 38.87 [37.02, 40.75] | 42.62 [40.54, 44.73] | -10.5 |
| | West | 32.07 [29.54, 34.70] | 51.24 [49.15, 53.33] | -4.0 |
| | South | 47.21 [44.68, 49.76] | 55.57 [51.59, 59.49] | -6.3 |

*(Continued)*

**Table 3.** (Continued)

| Background Characteristics | Non-communicable Disease | | |
|---|---|---|---|
| | Poor | Non-poor | Difference |
| Total (N = 65,562) | 26,131 | 39,431 | |

SC: Scheduled Caste, ST: Scheduled Tribe, OBC: Other Backward Caste; 95% confidence intervals are in brackets []

using nationally representative data. As discussed earlier, socio-economic inequality in relation to overall health has been studied innumerably in the previous literatures; but this study contributes in decomposing the socio-economic inequality in terms of monthly per capita consumption for communicable and non-communicable disease separately. This study significantly concludes, with the help of concentration index, its decomposition as well as through regression that, wealth or say economic status has highest contribution in creating inequality in disease prevalence among all other possible contributing factor.

In this study the communicable disease includes waterborne, vector borne and other infectious diseases and the non-communicable disease include CVD, Cancer, COPD, diabetes and bone disorder. This paper envisages the prevalence of both types of diseases among the older adult population and found that almost half of the older individuals suffer from NCD, and nearly one-fourth suffers from CD. The same pattern of NCD prevalence was discovered in a study based on wave one of the World Health Organization's Study on global ageing and adult health, which also indicated that 50 percent of the elderly population suffers from at least one type of chronic non-communicable illness [40]. Bivariate findings of the study also show a clear cluster of non-communicable disease among the non- poor class which is statistically proved through concentration index and concentration curve. The position of concentration curve supports that the concentration of communicable diseases is among the relatively poor strata where as the non-communicable disease is concentrated among non-poor strata. Among possible explanatory factors, unimproved source of drinking water along with other living environment, like shared and unimproved toilet are the major source of communicable disease spread. The study of Pathak [41], claimed that India had a higher prevalence of infectious diseases, particularly water bone disease as a result of a weak public drinking water distribution infrastructure, lends additional support to this. The result of logistic regression concludes that place of residence, educational status, BMI, type of house, type of toilet facility,

**Table 4. Association of consumption expenditure with communicable and non-communicable diseases.**

| MPCE Quintile | AOR (Adjusted Odds ratio) | 95% confidence interval |
|---|---|---|
| **Communicable disease** | | |
| Poorest® | | |
| Poorer | 0.18*** | [0.11,0.26] |
| Middle | 0.18*** | [0.11,0.25] |
| Richer | 0.24*** | [0.16,0.32] |
| Richest | 0.31*** | [0.22,0.41] |
| **Non-communicable disease** | | |
| Poorest® | | |
| Poorer | 1.20*** | [1.14,1.28] |
| Middle | 1.33*** | [1.25,1.41] |
| Richer | 1.52*** | [1.44,1.62] |
| Richest | 1.75*** | [1.65,1.86] |

**Table 5. Estimation of decomposition analysis for contribution of various explanatory variables for communicable diseases among older adults in India, 2017–18.**

| Explanatory Variables | Elasticity | Concentration Index (CI) | Absolute Contribution | Percentage Contribution |
|---|---|---|---|---|
| Age above 60 | -0.032 | -0.026 | 0.001 | -1.74 |
| Being Illiterate | 0.012 | -0.138 | -0.002 | 3.70 |
| Married | 0.018 | 0.013 | 0.000 | -0.44 |
| Muslim | -0.003 | 0.035 | 0.000 | 0.22 |
| Economic Status | -0.022 | -0.534 | 0.012 | -25.71 |
| Living with Spouse | 0.028 | 0.012 | 0.000 | -0.65 |
| Rural Residence | 0.061 | -0.118 | -0.007 | 15.69 |
| SC/ST | -0.017 | -0.180 | 0.003 | -6.54 |
| Female | 0.043 | 0.001 | 0.000 | 0.00 |
| Yoga | 0.004 | 0.179 | 0.001 | -1.53 |
| Physical Activity | 0.091 | -0.027 | -0.003 | 5.45 |
| Mobility Restrictions | 0.220 | -0.006 | -0.001 | 3.05 |
| ADL Problem | 0.030 | -0.019 | -0.001 | 1.31 |
| IADL Problem | 0.063 | -0.053 | -0.003 | 7.41 |
| Using Tobacco | 0.022 | -0.081 | -0.002 | 3.92 |
| Drinking Alcohol | -0.001 | -0.036 | 0.000 | 0.00 |
| Being Underweight | 0.018 | -0.209 | -0.004 | 8.06 |
| Being Overweight | -0.017 | 0.218 | -0.004 | 8.06 |
| Clean Cooking Fuel | 0.018 | -0.200 | -0.004 | 7.84 |
| Kutcha House | 0.002 | -0.287 | 0.000 | 0.87 |
| Not Improved Source of Drinking Water | 0.006 | 0.197 | 0.001 | -2.40 |
| Not Improved Toilet | 0.037 | -0.246 | -0.009 | 19.61 |
| North Region | 0.116 | 0.133 | 0.016 | -33.77 |
| Central Region | 0.204 | -0.133 | -0.027 | 59.26 |
| East Region | 0.092 | -0.153 | -0.014 | 30.72 |
| West Region | 0.047 | 0.027 | 0.001 | -2.83 |
| Northeast Region | 0.0046 | -0.0351 | -0.0002 | 0.44 |
| Residual | -0.001 | | | |

source of drinking water, and type of cooking fuel are important determinants of communicable disease among older adults in India. This aligns with the study of Dhara et al [42] that the infectious disease distribution which includes water borne diseases involves complex social and demographic factors including human population density and behaviour, housing type and location, water supply, sewage and waste management systems, land use and irrigation systems, access to health care, and general environmental hygiene. The study validates that age, sex, education, MPCE quintile, physical activities, mobility restriction, IADL restriction were significantly associated with non-communicable disease in the older population. The risk of having an NCD was found to be higher among respondents in the older age group; this conclusion is consistent with previous research's findings. The study by Syed et al [43] also revealed that the prevalence of NCDs showed an increasing trend with increasing age. The combined impact of several health issues and disease risk must be dealt with by older people due to the changing demographics and health situation in the nation. As a result, the country has a higher burden of NCDs among older adults [44–46]. According to the current study, people who live in urban areas have a higher burden of NCD; this conclusion is also confirmed by studies that use a person's place of residence as a predictor. According to these studies, the urban population's sedentary lifestyle is to blame for their increased risk of NCDs [43].

**Table 6. Estimation of decomposition analysis for contribution of various explanatory variables for non-communicable diseases among older adults in India, 2017–18.**

| Explanatory Variables | Elasticity | Concentration Index (CI) | Absolute Contribution | Percentage Contribution |
|---|---|---|---|---|
| Age above 60 | 0.105 | -0.026 | -0.003 | -3.45 |
| Being Illiterate | -0.048 | -0.138 | 0.007 | 8.57 |
| Married | -0.061 | 0.013 | -0.001 | -1.02 |
| Rural Residence | -0.103 | -0.118 | 0.012 | 15.47 |
| Living with Spouse | 0.052 | 0.012 | 0.001 | 0.77 |
| Economic Condition | -0.044 | -0.534 | 0.024 | 30.05 |
| Muslim | 0.009 | 0.035 | 0.000 | 0.38 |
| SC/ST | -0.015 | -0.180 | 0.003 | 3.45 |
| Female | -0.002 | 0.001 | 0.000 | 0.00 |
| Yoga | 0.014 | 0.179 | 0.003 | 3.20 |
| Physical Activity | -0.045 | -0.027 | 0.001 | 1.53 |
| Mobility Restrictions | 0.245 | -0.006 | -0.002 | -1.92 |
| ADL Problem | 0.025 | -0.019 | -0.001 | -0.64 |
| IADL Problem | 0.040 | -0.053 | -0.002 | -2.69 |
| Using tobacco | -0.010 | -0.081 | 0.001 | 1.02 |
| Drinking Alcohol | -0.010 | -0.036 | 0.000 | 0.51 |
| Being Underweight | -0.035 | -0.209 | 0.007 | 9.21 |
| Being Overweight | 0.073 | 0.218 | 0.016 | 20.46 |
| East Region | -0.028 | -0.153 | 0.004 | 5.50 |
| West Region | -0.015 | 0.027 | 0.000 | -0.51 |
| North Region | -0.002 | 0.133 | 0.000 | -0.38 |
| Northeast Region | -0.007 | -0.035 | 0.000 | 0.26 |
| Central Region | -0.060 | -0.133 | 0.008 | 10.23 |
| Residual | 0.094 | | | |

The results of this study show that the prevalence of non-communicable diseases varies with anthropometric status as determined by BMI level, with noticeably increased odds of prevalence among older persons who are overweight and obese compared to older adults with normal BMI. Similarly, the contribution of obesity in explaining inequality of non-communicable disease prevalence is also very high. The issue of Obesity was closely linked with an elevated risk of several major non-communicable diseases, including type 2 diabetes, coronary heart disease, stroke, asthma, and several cancers [47–49]. Moreover, the study found that the older adults from the Southern region were more likely to suffer from NCDs [45].

According to the study finding communicable disease was concentrated among poor older adults which is consistent with a study by Semenza and Giesecke [50] that found vulnerable populations are disproportionately afflicted by infectious diseases throughout all EU member states. Regional imbalance has highest contribution in explaining the health inequality for communicable disease; central, north and east region have highest contribution compared to south region. The possible reason for this regional inequality can be directly related to state level development and amenities index, the level and distribution of wealth within a society plays a significant role in determining vulnerabilities to communicable disease. On the other hand, the findings from the study depict an inclination of the non-communicable diseases (NCDs) among the respondents belonging to the richest wealth quintile which is further supported by the study of Sharma et al [51]. Similar findings have been seen in some low- and middle-income countries, where people from wealthy households are more likely to suffer from noncommunicable diseases [45, 46]. Our premise of exposure, accessibility, and

affordability, i.e., the fact that people from affluent communities are more exposed to learning about how people in the west lead their lifestyles, may be the main explanation for this conclusion. Additionally, they have access to these tools and have little trouble securing the financial resources needed to use them.

## 5. Conclusion

The study contributes crucially in policy making by dissecting the socio-economic inequality separately for communicable and non-communicable diseases. The findings can be applied to the entire nation because the study made use of data from a widely representative national large-scale survey. The study used data from a cross-sectional study therefore, the cross-sectional nature made it impossible to properly attribute causality and establish a relationship between explanatory variables and communicable and non-communicable diseases. Due to the study's reliance on self-reported data, there is also the potential for under- or over reporting of disease prevalence estimates, especially in poorer socio-economic group the lack of awareness and health screen regarding NCD can bias the self-reported prevalence.

## Supporting information

**S1 Appendix. Logistic regression estimates for older adults who suffered from communicable diseases by their background characteristics in India, 2017–18.**
(DOCX)

**S2 Appendix. Logistic regression estimates for older adults who suffered from non-communicable diseases by their background characteristics in India, 2017–18.**
(DOCX)

**S3 Appendix. Poor-rich ratio for communicable and non-communicable diseases among older adults across states in India, 2017–18.**
(DOCX)

## Acknowledgments

We would like to extent our gratitude to Arjun Jana for his sincere assistance in statistical analysis part.

## Author Contributions

**Conceptualization:** Jhumki Kundu, Ruchira Chakraborty.

**Data curation:** Jhumki Kundu, Ruchira Chakraborty.

**Formal analysis:** Jhumki Kundu, Ruchira Chakraborty.

**Methodology:** Ruchira Chakraborty.

**Software:** Ruchira Chakraborty.

**Validation:** Ruchira Chakraborty.

**Visualization:** Jhumki Kundu.

**Writing – original draft:** Jhumki Kundu, Ruchira Chakraborty.

**Writing – review & editing:** Jhumki Kundu, Ruchira Chakraborty.

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
