## [Decision Letter · Decision Letter 0]

17 Oct 2022

PONE-D-22-22415Socio-Economic Inequalities in Burden of Communicable and Non-Communicable Diseases Among Older Adults in India, 2017-18: Evidence from Longitudinal Ageing Study in India, 2017-18PLOS ONE

Dear Dr. Chakraborty,

Thank you for submitting your manuscript to PLOS ONE. After careful consideration, we feel that it has merit but does not fully meet PLOS ONE’s publication criteria as it currently stands. Therefore, we invite you to submit a revised version of the manuscript that addresses the points raised during the review process.

We look forward to receiving your revised manuscript.

Kind regards,

Innocent Ijezie Chukwuonye, MBBS, FMCP (Internal Medicine)

Academic Editor

PLOS ONE

2. We note you have included a table to which you do not refer in the text of your manuscript. Please ensure that you refer to Table 2 in your text; if accepted, production will need this reference to link the reader to the Table.

Reviewers' comments:

Reviewer's Responses to Questions

**Comments to the Author**

1. Is the manuscript technically sound, and do the data support the conclusions?

Reviewer #1: Partly

Reviewer #2: No

2. Has the statistical analysis been performed appropriately and rigorously? 

Reviewer #1: Yes

Reviewer #2: No

3. Have the authors made all data underlying the findings in their manuscript fully available?

Reviewer #1: No

Reviewer #2: Yes

4. Is the manuscript presented in an intelligible fashion and written in standard English?

Reviewer #1: No

Reviewer #2: No

5. Review Comments to the Author

Reviewer #1: Review of the paper “Socio-Economic Inequalities in Burden of Communicable and Non-Communicable Diseases Among Older Adults in India, 2017-18: Evidence from Longitudinal Ageing Study in India, 2017-18”

The study aimed to determine socioeconomic inequality in the burden of communicable and noncommunicable diseases among older adults in India. This study used Longitudinal Ageing study in India (LASI), Wave1, conducted during 2017–2018.

This is an important and interesting topic to policy makers, but several aspects of this paper need to be improved.

Major issues

1. In the Abstract and Variables paragraphs, isn’t clear if Communicable and NCD are considered in the same explanatory variable or separately. You understand that are estimated as two separate models only when you read table’s results. It’s recommended to clarify this aspect in the Abstract section.

2. The Figure 1, showing conceptual framework, doesn’t help to understand relationship between communicable and NCD diseases that are treated as the same variable with the same determinants. Add a note to explain outcome, discriminate for determinant in CD and NCD or delete the figure 1 because it isn’t recalled in any part of the paper.

3. Table 1 has some calculation errors in several rows: the total is not 100 as it should be. Check and correct all values in the table

4. Tables 3 and 4 lack references on statistical test considered. What does the test measure, the differences or the percentages between poor and not poor? Clarify and correct this table

5. Table 4 has some errors in difference between poor and non-poor (one example for age group45-59, 33.5-41.9=-8.4 not 8.5). Check and correct all the values.

6. Table 5 shows that age isn’t significative, how do you explain this result? Not is usual to observe not significance between age and health outcome. Add the reasons and references to lead this result

7. With reference to Tables 7 and 8, the authors should clarify how absolute and percentage contributions were calculated

8. In the Discussion, authors assert that their study “uniquely contributes in decomposing the socio-economic inequality in terms of communicable and non-communicable disease separately…” but this topic was already treated in different studies. At European level, for example I suggest to read the recent paper “Bono F. & Matranga D (2019) Socioeconomic inequality in non-communicable diseases in Europe between 2004 and 2015: evidence from SHARE survey”, published in the European journal of Public Health, that carried out an interesting analysis comparing different European countries in terms of inequality of non-communicable diseases, highlights on the determinants of inequality and its decomposition. Results show that among socioeconomic determinants, education and marital status are the most concentrated and the inequality is attributed mostly to physical inactivity and obesity and this contribution increased during the study period.

Minor issues

The text has several formatting errors, authors must carefully read the paper.

For example several paragraph began with number: at page 19 there are several of these type of errors.

At page 9, the sentence “Meanwhile, the burden of non-communicable diseases…”.is incomplete or incorrect

At page 19, at the middle of the page ”Olde adult….” Correct

At page 21, the sentence” For communicable disease (Table 3), with…” isn’t clear but even if it had been clear will be incorrect because age isn’t significative

Reviewer #2: The authors present a descriptive analysis of some communicable and non-communicable diseases among the Indian population, ages 45 and above, across a variety of demographic characteristics with a particular emphasis on correlation between disease incidence and socioeconomic inequality. This analysis differs from others in that considers communicable diseases separately from noncommunicable diseases. The data analyzed are a one-year snapshot (2017-2018) from the Longitudinal Ageing Study in India (LASI).

While the premise of this analysis is interesting, the manuscript was written in such a way as to make it difficult to understand at times. I encourage the authors to seek independent editorial help before submitting a revision. At present it is difficult to offer substantial recommendations for revision because at times I was unable to follow the positions being advanced. That said, I present some general observations below:

1. Background

Paragraph 2: "Due to the often-fragile healthcare systems' [sic] lack of funding..." Which health care systems are being described here? The authors later specify low-income and middle-income countries, but of the two articles they cite, one is specifically about Kyrgyzstan--a single country--and the other is specifically about major infectious diseases control priorities.

2.2.1 Outcome Variables

The authors state that an outcome variable of "1" is assigned for observations where either communicable or non-communicable disease was indicated, otherwise it was coded as "0". In other words, the control variable indicates the presence or absence of disease, regardless of type. However, these two conditions appear to be considered separately throughout the rest of the paper.

It is not clear exactly which diseases are included among "communicable disease" and which are included among "non-communicable disease". The authors state, for example, that the communicable diseases category "includes diseases that are diarrhoea[sic]/gastroenteritis/typhoid/jaundice/hepatitis,[sic]malaria/chikungunya/dengue and other infectious diseases..." but it is unclear what else this category contains (or excludes). I would also note that jaundice itself is not contagious/communicable although some of the underlying conditions associated with jaundice can be.

2.2.2 Explanatory variables

The authors not that "control variables were selected after doing extensive literature review" but the majority of proposed variables do not contain references. Their justification is otherwise unstated an is therefore unclear.

3.5 Prevalence of communicable and non-communicable disease in different socioeconomic background

The authors state that "Weighted prevalence was calculated for separate sets of explanatory variables for both the group of disease" but the relevant tables--Table 3 and Table 4--do not appear to show weighted prevalence results. If weighted prevalence was indeed calculated, please explain the calculation and report confidence intervals on the relevant tables.

4 Discussion

In paragraph 3, the authors state that "The combined impact of several health issues and disease risk must be dealt with by older people due to the changing demographics and health situation in the nation. As a result, the country has a higher burden of NCDs among older adults [41,42,43]" It is not persuasive. Is it not true that increased age is a specific risk factor in many non-communicable diseases, such as cardiovascular disease? In addition, the sentence is written in such a way as to imply that the references are about India, however only reference 41 focuses on India. The other two focus on sub-Saharan Africa and South Africa specifically.

6. PLOS authors have the option to publish the peer review history of their article (what does this mean?). If published, this will include your full peer review and any attached files.

Reviewer #1: No

Reviewer #2: No

---

## [Author Response · Author response to Decision Letter 0]

14 Dec 2022

Dear Editor, 

Thank you for giving us the opportunity to submit a revised draft of the manuscript titled “Socio-econmic Inequalities in Burden of Communicable and Non-communicable Diseases among Older Adults in India: Evidence from Longitudinal Ageing Study in India, 2017-18.” in your esteemed journal. We have modified the manuscript according to the suggestion given by you and all the reviewers. Here we have tried to address all the issues raised. 

Point by Point Response

Editor’s Comment: We note you have included a table to which you do not refer in the text of your manuscript. Please ensure that you refer to Table 2 in your text; if accepted, production will need this reference to link the reader to the Table.

Author’s Response: We have cited the aforesaid table in the revised manuscript. 

Major issues

Reviewer’s Comment: In the Abstract and Variables paragraphs, isn’t clear if Communicable and NCD are considered in the same explanatory variable or separately. You understand that are estimated as two separate models only when you read table’s results. It’s recommended to clarify this aspect in the Abstract section.

Author’s Response: In our manuscript we have analysed the socio-economic inequalities of communicable and non-communicable diseases among older adults in India with separate sets of explanatory variables. As per your suggestion, in abstract, the line has been modified as ‘chosen set of separate explanatory variables’. In the variables paragraphs also, the explanatory variables which are common for both communicable and non-communicable diseases are mentioned separately and the variables unique for communicable and noncommunicable diseases are mentioned separately. 

Reviewer’s Comment: The Figure 1, showing conceptual framework, doesn’t help to understand relationship between communicable and NCD diseases that are treated as the same variable with the same determinants. Add a note to explain outcome, discriminate for determinant in CD and NCD or delete the figure 1 because it isn’t recalled in any part of the paper.

Author’s Response: As a diagrammatic representation of conceptual framework provides a comprehensive understanding we opted for not to delete the figure but we have modified the figure completely as per your suggestion where both the outcome variables along with their common and separate sets of explanatory variables are shown separately. 

Reviewer’s Comment: Table 1 has some calculation errors in several rows: the total is not 100 as it should be. Check and correct all values in the table

Author’s Response: As per the comment we have corrected the calculation errors.

Reviewer’s Comment: Tables 3 and 4 lack references on statistical test considered. What does the test measure, the differences or the percentages between poor and not poor? Clarify and correct this table

Author’s response: In the revised manuscript we have added the confidence interval for all the prevalence presented in both Table 3 and Table 4. Thus, it could be clearly identified whether the difference between poor and non-poor is statistically significant or not. 

Reviewer’s Comment: Table 4 has some errors in difference between poor and non-poor (one example for age group45-59, 33.5-41.9=-8.4 not 8.5). Check and correct all the values.

Author’s response: As per the comment we have corrected the values in Table 4.

Reviewer’s Comment: Table 5 shows that age isn’t significative, how do you explain this result? Not is usual to observe not significance between age and health outcome. Add the reasons and references to lead this result

Author’s response: Table 5 showed the Logistic regression estimates for older adults who suffered from communicable diseases by their background characteristics in India, 2017-18. In our analysis communicable disease included waterborne disease (diarrhea/gastroenteritis, typhoid, jaundice or hepatitis), vector borne disease (malaria, dengue or chikungunya) and other infectious disease (tuberculosis and urinary tract infection). Age does not appear to be a significant determinant for communicable diseases in the logistic regression estimations. In table 3 as well we can see that prevalence of communicable disease does not significantly vary between the age groups as the confidence interval collides. Thus, our findings align in same direction with all the analysis. 

Age was found to be a significant determinant for non-communicable diseases in the majority of the research, but not for communicable diseases. Similarly, age does not appear a significant variable in the study of kumar et al, 2022 “Prevalence and predictors of water-borne diseases among elderly people in India: evidence from Longitudinal Ageing Study in India, 2017-18”, which we have given reference in the previous manuscript.

Although the increased susceptibility of older persons to infectious/communicable diseases frequently has been attributed to the decline in immune function that occurs with aging, there are very few data confirming this hypothesis.

Reviewer’s Comment: With reference to Tables 7 and 8, the authors should clarify how absolute and percentage contributions were calculated

Author’s response: Absolute contribution of each determinants is calculated by multiplying the elasticity and the CI of each control variable and the percentage contribution is calculated by dividing the absolute contribution with overall CI. This has been added in statistical analysis section of the revised manuscript.

Reviewer’s Comment: In the Discussion, authors assert that their study “uniquely contributes in decomposing the socio-economic inequality in terms of communicable and non-communicable disease separately…” but this topic was already treated in different studies. At European level, for example I suggest to read the recent paper “Bono F. & Matranga D (2019) Socioeconomic inequality in non-communicable diseases in Europe between 2004 and 2015: evidence from SHARE survey”, published in the European journal of Public Health, that carried out an interesting analysis comparing different European countries in terms of inequality of non-communicable diseases, highlights on the determinants of inequality and its decomposition. Results show that among socioeconomic determinants, education and marital status are the most concentrated and the inequality is attributed mostly to physical inactivity and obesity and this contribution increased during the study period.

Author’s response: In the discussion what we have tried to mention is, this research tried to provide a contemporary picture of the scenario in inequality of health in terms of both communicable and non-communicable diseases specifically in Indian scenario. The research article by Bono F. & Matranga D (2019), only dealt with non-communicable diseases not communicable and non-communicable together to compare and that too in European scenario. In that way, we have put the sentence that our study contributes in identifying factors in Indian context, but as per your suggestion we have omitted the word ‘uniquely’. 

Minor issues

Reviewer’s Comment: The text has several formatting errors, authors must carefully read the paper.

For example several paragraphs began with number: at page 19 there are several of these types of errors.

Author’s response: We have tried to correct the typographical errors we could find in the revised manuscript. 

Reviewer’s Comment: At page 9, the sentence “Meanwhile, the burden of non-communicable diseases…”.is incomplete or incorrect

Author’s response: We looked at the text "Meanwhile, the burden of non-communicable diseases..." in light of your opinion. As far as we are aware, the statement is complete, and we have already cited it.

Meanwhile, the burden of non-communicable diseases (NCD), such as diabetes, cancer, and cardiovascular illnesses, has increased quickly as a result of the change in nutritional and lifestyle behavioural patterns (such as increasing fast food consumption, alcohol intake, and tobacco use) [Boutayeb & Boutayeb ,2005, Cappuccio et al.,2016 & Schwartz et al.,2015].

Reviewer’s Comment: At page 19, at the middle of the page “Olde adult….” Correct

Author’s response: The typographical error is corrected in the revised manuscript. 

Reviewer’s Comment: At page 21, the sentence” For communicable disease (Table 3), with…” isn’t clear but even if it had been clear will be incorrect because age isn’t significative

Author’s response: The sentence has been omitted in the revised manuscript. 

Reviewer #2: The authors present a descriptive analysis of some communicable and non-communicable diseases among the Indian population, ages 45 and above, across a variety of demographic characteristics with a particular emphasis on correlation between disease incidence and socioeconomic inequality. This analysis differs from others in that considers communicable diseases separately from noncommunicable diseases. The data analyzed are a one-year snapshot (2017-2018) from the Longitudinal Ageing Study in India (LASI).

While the premise of this analysis is interesting, the manuscript was written in such a way as to make it difficult to understand at times. I encourage the authors to seek independent editorial help before submitting a revision. At present it is difficult to offer substantial recommendations for revision because at times I was unable to follow the positions being advanced. That said, I present some general observations below:

1.Background

Paragraph 2: "Due to the often-fragile healthcare systems' [sic] lack of funding..." Which health care systems are being described here? The authors later specify low-income and middle-income countries, but of the two articles they cite, one is specifically about Kyrgyzstan--a single country--and the other is specifically about major infectious diseases control priorities.

Author’s response: The line is mentioned in the background section of the manuscript to provide a wholesome picture of disease burden according to socio-economic characteristics. As the research is based on Indian sample, India also comes under a lower-middle income country and most of these countries suffers from similar health system problems; like lack of funding in health administration to control communicable disease burden. And the two references mentioned here are about lower- and middle-income countries, that is why the line has been kept intact. 

Reviewer’s Comment- 2.2.1 Outcome Variables

The authors state that an outcome variable of "1" is assigned for observations where either communicable or non-communicable disease was indicated, otherwise it was coded as "0". In other words, the control variable indicates the presence or absence of disease, regardless of type. However, these two conditions appear to be considered separately throughout the rest of the paper.

Author’s response: In our manuscript we have mentioned two outcome variables; “Communicable” & “Non-communicable disease”, because our purpose was to assess the socio-economic inequalities for both communicable & non-communicable diseases among older adults in India. Two separate logistic regression models were estimated for communicable and non-communicable disease. In table 5 we did a logistic regression estimates for older adults who suffered from communicable diseases by their background characteristics in India, where the outcome variable was communicable disease which was coded as “1” for having the communicable disease and “0” for not having the communicable disease, and in Table 6 we did a logistic regression estimate for older adults who suffered from non-communicable diseases by their background characteristics in India, where the outcome variable was non-communicable disease which was coded as “1” for having the non-communicable disease and “0” for not having the non-communicable disease.We have estimated the prevalence of both communicable & Non-Communicable Disease according to socio-economic Characteristics in India. For measurement of socioeconomic inequality, concentration curve and concentration index along with state wise poor-rich ratio was calculated for both communicable and non-communicable diseases. We have Estimated the decomposition analysis for contribution of various explanatory variables for both communicable and non-communicable diseases among older adults in India.

Reviewer’s Comment: It is not clear exactly which diseases are included among "communicable disease" and which are included among "non-communicable disease". The authors state, for example, that the communicable diseases category "includes diseases that are diarrhoea[sic]/gastroenteritis/typhoid/jaundice/hepatitis, [sic]malaria/chikungunya/dengue and other infectious diseases..." but it is unclear what else this category contains (or excludes). I would also note that jaundice itself is not contagious/communicable although some of the underlying conditions associated with jaundice can be.

Author’s Response: In our manuscript we have mentioned that other infectious diseases included tuberculosis/ urinary tract infection. 

The Longitudinal Ageing study in India (LASI) survey collected the information on communicable disease for diarrhoea/gastroenteritis/ typhoid/ jaundice/hepatitis, malaria/ chikungunya/ dengue and other infectious diseases like tuberculosis/ urinary tract infection, so in our analysis we have only included these diseases under the communicable disease category. As a result, we are unable to specify which other diseases were not considered under the communicable disease category.

 In the LASI survey jaundice was considered as waterborne disease so we have included jaundice as a communicable disease in our analysis.

Reviewer’s Comment: 2.2.2 Explanatory variables

The authors not that "control variables were selected after doing extensive literature review" but the majority of proposed variables do not contain references. Their justification is otherwise unstated an is therefore unclear.

Author’s Response: We have mentioned the references of some of explanatory variables like type of toilet facility which was recoded as unimproved and improved [Kumar et al.,2021], Source of drinking water which was recoded as unimproved and improved [Kumar et al.,2021], Body mass index was recoded as underweight, normal and overweight/obese [Fauziana et al.,2016].

 As per the comment we have added the references of caste category [Zacharias et al. Kumar et al.,2022], Types of house category (pucca, semi pucca and kutcha) [Kumar et al.,2022], and alcohol consumption status [ Bose et al.,2019]

Reviewer’s Comment: 3.5 Prevalence of communicable and non-communicable disease in different socioeconomic background-

The authors state that "Weighted prevalence was calculated for separate sets of explanatory variables for both the group of disease" but the relevant tables--Table 3 and Table 4--do not appear to show weighted prevalence results. If weighted prevalence was indeed calculated, please explain the calculation and report confidence intervals on the relevant tables.

Author’s Response: The above-mentioned tables do show weighted prevalence of communicable and non-communicable disease separately. LASI data set provides two separate weights, national level individual weights and state level weights. In this study for table 3 and table 4 we have used India level weight as we are calculating national level estimates. As per the comment we have added the confidence interval in both the tables. 

Reviewer’s Comment: 4 Discussion

In paragraph 3, the authors state that "The combined impact of several health issues and disease risk must be dealt with by older people due to the changing demographics and health situation in the nation. As a result, the country has a higher burden of NCDs among older adults [41,42,43]" It is not persuasive. Is it not true that increased age is a specific risk factor in many non-communicable diseases, such as cardiovascular disease? In addition, the sentence is written in such a way as to imply that the references are about India, however only reference 41 focuses on India. The other two focus on sub-Saharan Africa and South Africa specifically.

Author’s response: We completely agree with you on the point and it is also reflected in our analysis as well that with increasing age the likelihood of having non-communicable disease increases manifold. The line we used in discussion tried to elaborate this concept that as India is ageing at a very faster pace, with changing demographic situation, health situation will also change; the burden of non-communicable disease will increase automatically due to the aforementioned fact. And in different literatures in population studies suggests similar condition of Sub-Saharan Africa and Indian ageing scenario though they are at different stages of demographic transition. That is why we have tried to mentioned Indian scenario as well as cited similar studies from African scenario.

---

## [Editor Report · Decision Letter 1]

12 Jan 2023

PONE-D-22-22415R1Socio-economic Inequalities in Burden of Communicable and Non-communicable Diseases among Older Adults in India: Evidence from Longitudinal Ageing Study in India, 2017-18PLOS ONE

Dear Dr. Chakraborty,

Thank you for submitting your manuscript to PLOS ONE. After careful consideration, we feel that it has merit but does not fully meet PLOS ONE’s publication criteria as it currently stands. Therefore, we invite you to submit a revised version of the manuscript that addresses the points raised during the review process. The majority of the reviewers' concerns have been addressed; however, the article still has grammatical errors.

In my opinion, you should employ an English language expert to edit the article.

Furthermore, some of your references did not adhere to PLOS ONE's journal guidelines; please review them to make sure they conform to the journal's standards.

We look forward to receiving your revised manuscript.

Kind regards,

Innocent Ijezie Chukwuonye, MBBS, FMCP (Internal Medicine)

Academic Editor

PLOS ONE
---

## [Author Response · Author response to Decision Letter 1]

4 Mar 2023

Dear Editor,

 Thank you for giving us the opportunity to revise the article. We have addressed all the raised pointed below. 

Editor’s comment: We've checked your submission and before we can proceed, we need you to address the following issues:

1. We notice that your manuscript file was uploaded on December 14, 2022. Please can you upload the latest version of your revised manuscript as the main article file, ensuring that does not contain any tracked changes or highlighting. This will be used in the production process if your manuscript is accepted. Please follow this link for more information: http://blogs.PLOS.org/everyone/2011/05/10/how-to-submit-your-revised-manuscript/

Author’s Reply: Thank you for pointing out the mistake. The updated manuscript has been uploaded in the portal. 

Editor’s comment: Please ensure that you refer to Figures 2 and 6 in your text as, if accepted, production will need this reference to link the reader to the figure.

Author’s Reply: We have thoroughly checked the manuscript and reference text for figure 2 was already present in the manuscript, however there is no figure 6 uploaded so there was nothing to mention about figure 6. We have only uploaded figure 1-5. 

Editor’s comment: Please upload a copy of Figures 1 and 7, and 8 which you refer to in your text. Or if the figure is no longer to be included as part of the submission please remove all reference to it within the text. 

Author’s Reply: Copy of figure 1 was already uploaded and checked in the pdf builder. There was no figure 7 and 8, neither we have mentioned that in our text, we have checked that thoroughly. The figures we have uploaded is restricted to figure 1- figure 5.

---

## [Editor Report · Decision Letter 2]

7 Mar 2023

Socio-economic Inequalities in Burden of Communicable and Non-communicable Diseases among Older Adults in India: Evidence from Longitudinal Ageing Study in India, 2017-18

PONE-D-22-22415R2

Dear Ruchira Chakraborty

We’re pleased to inform you that your manuscript has been judged scientifically suitable for publication and will be formally accepted for publication once it meets all outstanding technical requirements.

Kind regards,

Innocent Ijezie Chukwuonye, MBBS, FMCP (Internal Medicine)

Academic Editor

PLOS ONE

---

## [Editor Report · Acceptance letter]

20 Mar 2023

PONE-D-22-22415R2 

Socio-economic Inequalities in Burden of Communicable and Non-communicable Diseases among Older Adults in India: Evidence from Longitudinal Ageing Study in India, 2017-18 

Dear Dr. Chakraborty:

I'm pleased to inform you that your manuscript has been deemed suitable for publication in PLOS ONE. Congratulations! Your manuscript is now with our production department. 

Kind regards, 

on behalf of

Dr. Innocent Ijezie Chukwuonye 

Academic Editor

PLOS ONE